



**The linear growth rate of Rayleigh-Taylor instability in ionospheric F layer**

3                                  Kangkang Liu[1,2,*]

[1] Key Laboratory of Earth and Planetary Physics, Institute of Geology and Geophysics, Chinese Academy
of Sciences, Beijing, China.
[2] University of Chinese Academy of Science, Beijing, China.
* Corresponding author: Kangkang Liu (zongkangliu@163.com)
**Abstract**
It is generally considered that the perturbation electric field generated by the charge
accumulation caused by the current divergence is the driving force for Rayleigh-Taylor
instability (RTI) in plasma. However, in previous calculation of the linear growth rate
of RTI the current continuity equation was applied, which means the contribution of
charge accumulation to the growth of RTI was ignored. Applying the perturbation
electric field and the current continuity equation simultaneously in calculating the linear
growth rate of RTI of the ionospheric F layer will give erroneous results. In this paper,
we calculated the linear growth rate of RTI with the standard instability analysis method.
The charge conservation equation was used in the calculation instead of the current
continuity to study the contribution of charge accumulation to the growth of RTI. The
results show that the contribution of charge accumulation to the linear growth rate of
RTI is proportional to the ratio of Alfvén speed to the light speed. In ionospheric F layer
the ratio is small, the contribution of charge accumulation to the growth of RTI is
negligible. This indicates that the previous physical description of the RTI in the
ionospheric F layer is wrong and a new physical description of RTI is needed. In the
new physical description perturbation electric field and charge accumulation is not the
cause, but the result of RTI. In ionospheric layer, background electric field and neutral
wind velocity have no effect on the linear growth rate of RTI.

Keywords: Rayleigh-Taylor instability; equatorial plasma bubble; linear growth rate





## 1 Introduction

When a heavier fluid is supported by a lighter fluid against gravity, the equilibrium is unstable to any perturbations of the interface. For if a parcel of heavier fluid is displaced downward with an equal volume of lighter fluid displaced upwards, the potential energy of the configuration is lower than that in the initial state, and the process goes on. This instability is called the Rayleigh-Taylor Instability (RTI) (Taylor, 1950). In addition to neutral fluids, RTI also plays an important role in space and laboratory plasmas (Isobe et al., 2005;Sultan, 1996;Robinson et al., 2004;Atzeni et al., 2004;Ryutov et al., 2000).

Unlike in neutral fluids, heavier fluids are supported by lighter fluids in equilibrium state. In the plasma, in the equilibrium state, the heavier plasma is also partially supported by the magnetic field. The intuitive physical description of RTI in ionospheric F layer is shown in figure 1 (Kelley, 2009). In equilibrium state a net current flows in in the horizontal direction and the current is proportional to the plasma density. There is thus a divergence, and charge will pile up on the edges of the small initial perturbation. As a result, perturbation electric fields build up in the directions shown. These fields in turn cause an upward (downward) drift in the region where the density is low (high). Lower (higher) density plasma is therefore advected upward (downward), creating a larger perturbation, and the system is unstable. In the above physical description charge accumulation is the cause for the growth of RTI. However, in the calculation of the linear growth rate of RTI, the current continuity equation is applied (Sultan, 1996;Chandrasekhar, 2013;Sharp, 1983). Form the charge conservation equation $\nabla \cdot \boldsymbol{J} + \frac{\partial \rho}{\partial t} = 0$, we know that when current continuity equation $\nabla \cdot \boldsymbol{J} = 0$ is applied, there will be not charge accumulation due to the divergence of the current. Therefore, the contribution of charge accumulation to the growth of RTI is ignored, and this contradict with the intuitive physical description of RTI in figure 1.

The linear growth rate of RTI in ionospheric F layer calculated by Kelley (2009) is $\gamma = \frac{g}{L v_{in}}$, which can explain many statistical characteristics of equatorial plasma bubble (EPB) and was widely used in the EPB literature (Yokoyama, 2017, and references therein). However, in their calculation perturbation electric field and current continuity equation were applied at the same time. It should be noted that when current continuity equation applied, there will be no perturbation electric field due to charge accumulation. The linear growth rate he calculated was not accurate. For example, the linear growth rate tends to infinity when the collision frequency approaches zero. The linear growth rate of RTI when collision frequency approaches zero should reduce to that of magnetized plasma without neutral particles $\gamma = \sqrt{\frac{g}{L}}$.

In this paper, we calculated the linear growth rate of RTI with the standard instability analysis method. The effects of different factors on the linear growth rate of RTI were discussed. A new expression of linear growth rate is calculated and a new physical



description of RTI in ionospheric F layer was depicted.
**2 Mathematical model and dispersion relation**
Assuming incompressible plasma is composed of two kinds of particles, with $m_e \ll$
$m_i$, where $m_e$ and $m_i$ are the electron mass and ion mass, respectively. The plasma
and homogeneous neutral wind are immersed in magnetic field B (0, 0, B) and gravity
field g (0,-g, 0) (see figure 1). Ignore the electron collision and $m_e/m_i$ terms (Kelley,
2009).The relevant equations in the Gaussian unit can be written as:
$$\frac{\partial(\rho V)}{\partial t} = \frac{1}{c}J \times B + \rho g - \nabla p - \rho v_{in}(V - V_n) \tag{1}$$
$$\frac{\partial E}{\partial t} = -4\pi J + c\nabla \times B \tag{2}$$
$$\frac{\partial B}{\partial t} = -c\nabla \times E \tag{3}$$
$$\nabla \cdot B = 0 \tag{4}$$
$$\nabla \cdot E = 4\pi \rho_c \tag{5}$$
$$\nabla \cdot J = -\frac{\partial \rho_c}{\partial t} \tag{6}$$
$$\frac{\partial \rho}{\partial t} + \nabla \cdot (\rho V) = 0 \tag{7}$$
Where $V$, $\rho$ , $c$, $J$, $B$, $g$, $p$, $E$, $\rho_c$, $V_n$, $v_{in}$ are the velocity of plasma fluid
element, plasma mass density, light speed, electric current density, magnetic field,
gravity acceleration, thermal pressure, electric field, charge density, neutral wind
velocity, ion-neutral collision frequency, respectively.
To examine the stability of the system, we assume the following perturbation in
physical quantities
$\rho = \rho^0 + \rho^1$, $p = p^0 + p^1$, $B = B^0 + B^1$, $J = J^0 + J^1$, $V = V^0 + V^1$, $V^0=0$,
$E = E^0 + E^1$, $E^0=0$.
Assuming perturbations in the form
$$\psi \propto \psi(y)e^{i(kx-\omega t)} \tag{8}$$
where $\omega$ is the frequency of the perturbation, $k$ is the wave number.
Linearizing the Eq. (1), we get
$$\rho^0 \frac{\partial V}{\partial t} = \frac{1}{c}J^0 \times B^1 + \frac{1}{c}J^1 \times B^0 + \rho^1 g - \nabla p^1 - \rho^0 v_{in}V \tag{9}$$
$z \cdot \nabla \times$ Eq. (9) yields
$$-i\omega(ik\rho^0 V_y - \frac{\partial}{\partial y}(\rho^0 V_x)) = -ik\rho^1 g - \frac{1}{c}(\nabla \cdot J^1)B^0 - v_{in}(ik\rho^0 V_y - \frac{\partial}{\partial y}(\rho^0 V_x)) \tag{10}$$
where $z$ is the unit vector in the $z$-direction.
From the assumption that the plasma is incompressible
$$\nabla \cdot V = 0 \tag{11}$$
We get the following equation
$$V_x = \frac{i}{k}\frac{\partial V_y}{\partial y} \tag{12}$$



From the continuity equation
$\frac{\partial \rho^1}{\partial t} + \boldsymbol{V} \cdot \nabla \rho^0 = 0$ (13)
We get
$\rho^1 = \frac{1}{iw}\frac{\partial \rho^0}{\partial y}V_y$ (14)
From Eq. (5) and Eq. (6), we get
$\nabla \cdot \boldsymbol{J}^1 = -\frac{\partial \rho_c}{\partial t} = \frac{\partial \rho_c b}{\partial t} = \frac{1}{4\pi}\nabla \cdot \left(\frac{\partial \boldsymbol{E}}{\partial t}\right)$ (15)
where $\rho_c$ $(\rho_c b)$ is the charge accumulation in (outside) the fluid element. This term
estimates the contribution of charge accumulation to the growth rate of RTI.

The exact relation between **E** and **v** in collisional plasma is not simply $c\boldsymbol{E} + \boldsymbol{V} \times \boldsymbol{B} =$
0, From the generalized Ohm's law (Vasyliunas, 2005), we get that with the given **E**
the maximum **v** is given by the above relation. For simplicity we will use the above
relation and note that the contribution of charge accumulation to the growth of RTI is
maximized. From the above relation we get
$E_x = -\frac{1}{c}V_y B^0$ (16)
Substituting Eq. (12), Eq. (14) and Eq. (16) into the Eq. (10), we get
$(\omega\rho^0 + iv_{in}\rho^0)\frac{\partial^2 V_y}{\partial y^2} + (\omega\frac{\partial \rho^0}{\partial y} + iv_{in}\frac{\partial \rho^0}{\partial y})\frac{\partial V_y}{\partial y} - k^2(\omega\rho^0 + iv_{in}\rho^0 + \frac{g}{\omega}\frac{\partial \rho^0}{\partial y} -$
$\frac{1}{4\pi}\frac{B^{0^2}}{c^2}\omega)V_y = 0$ (17)
Supposing the initial plasma density has an exponential dependence on $y$, that is
$\rho^0(y) = \rho^0 e^{\frac{y}{L}}$ (18)
Where $L$ is the gradient scale length. Substituting Eq. (18) into the equation (17) we get
$(\omega + iv_{in})\frac{\partial^2 V_y}{\partial y^2} + \frac{1}{L}(\omega + iv_{in})\frac{\partial V_y}{\partial y} - k^2(\omega + iv_{in} + \frac{g}{L\omega} - \frac{V_A^2}{c^2}\omega)V_y = 0$ (19)
where $V_A^2 = \frac{B^{0^2}}{4\pi\rho^0}$ is the square of the Alfvén speed.
Supposing that the stratified plasma of finite thickness is bounded by two rigid
boundaries $y = 0$ and $y = h$, the discrete solutions of Eq. (19) can be found of the form
$V_y(y) = C_0 \sin(\frac{m\pi y}{h})e^{-\frac{y}{2L}}$ (20)
Where $C_0$ is a constant. Substituting the Eq. (20) into the equation (19), we get a
general dispersion relation
$\omega = \frac{-iv_{in}D_1 \pm i\sqrt{v_{in}^2 D_1^2 + 4(D_1 - D_2)*D_3}}{2(D_1 - D_2)}$ (21)
Where
$D_1 = \frac{1}{4L^2} + \frac{m^2\pi^2}{h^2} + k^2$ (22)





$D_2 = k^2 \dfrac{V_A{}^2}{c^2}$                                                                  (23)
$D_3 = k^2 \dfrac{g}{L}$                                                                   (24)

### 3 The impact of various factors on the linear growth rate of RTI

Form Eq. (21) we know that the linear growth rate of RTI is
$\gamma = \dfrac{-v_{in}D_1 + \sqrt{v_{in}{}^2 D_1{}^2 + 4(D_1 - D_2)*D_3}}{2(D_1 - D_2)}$                  (25)

#### 3.1 Absence of collision and charge accumulation

When $v_{in} = 0$ and $V_A{}^2 = 0$ the growth rate reduces to
$\gamma = \dfrac{\sqrt{4 D_1 * D_3}}{2 D_1} = \left(\dfrac{g}{L} \dfrac{h^2 k^2}{h^2 k^2 + m^2 \pi^2 + h^2/4 L^2}\right)^{\frac{1}{2}}$             (26)
This is the growth rate of classical RTI. With k increases, the growth rate tends to a
maximum value
$\gamma = \sqrt{\dfrac{g}{L}}$                                                                 (27)

#### 3.2 Influence of charge accumulation on the linear growth rate of RTI

When $v_{in} = 0$ the growth rate is
$\gamma = \dfrac{\sqrt{4(D_1 - D_2)*D_3}}{2(D_1 - D_2)} = \left(\dfrac{g}{s} \dfrac{h^2 k^2}{h^2 k^2 + m^2 \pi^2 + h^2/4 L^2 - V_A{}^2/c^2}\right)^{\frac{1}{2}}$    (28)
To investigate the effect of charge accumulation on the linear growth rate of RTI, we
normalize equation (29) with the following expressions
$\gamma^* = \gamma(\omega_{pe})^{-1}, \; g^* = g(L\omega_{pe}^2)^{-1}, \; k^* = kL, \; h^* = h(L)^{-1}$
Where $\omega_{pe}$ is the plasma frequency. We get
$\gamma^* = \left(g^* \dfrac{h^{*2} k^{*2}}{h^{*2} k^{*2} + m^2 \pi^2 + h^{*2}/4 - V_A{}^2/c^2}\right)^{1/2}$            (29)
Figure 2 shows the dimensionless dispersion relation for configuration, $h^* = 1$, $m = $
$1$, $g^* = 10$, $g^* = 10$, $V_A{}^2/c^2 = 0$, 0.01, 0.1, 0.3, 0.5. Note that the curve representing
$V_A{}^2/c^2 = 0.01$ is basically coincides with that of $V_A{}^2/c^2 = 0$. When $V_A = 0$, equation (28)
represents the dispersion relation for the classical RTI (Goldston and Rutherford, 1995),
and the growth rate is the same as that of the classical RTI. When $V_A > 0$, the growth
rate is larger than that of the classical RTI and increases with the increase of $V_A{}^2/c^2$. It
can be seen from figure 2 that only when $V_A$ is large will charge accumulation have a
significant effect on the growth rate of RTI, otherwise, the effect can be neglected.

#### 3.3 Influence of collision frequency on the linear growth rate of RTI

Ignore the $V_A{}^2/c^2$ term, the growth rate reduce to



$$\gamma = \frac{-v_{in}D_1 + \sqrt{v_{in}^2 D_1^2 + 4D_1 * D_3}}{2D_1} \tag{30}$$
With k increases, the growth rate tends to a maximum value
$$\gamma = \sqrt{\frac{g}{L} + \frac{v_{in}^2}{4}} - \frac{v_{in}}{2} \tag{31}$$
Normalize Eq. (30) with the following expressions
$\gamma^* = \gamma(\omega_{pe})^{-1}, \ g^* = g(L\omega_{pe}^2)^{-1}, \ k^* = kL \ , \ h^* = h(L)^{-1}, \ v_{in}^* = v_{in}(\omega_{pe})^{-1}$
We get
$$\gamma^* = -\frac{v_{in}^*}{2} + \frac{\sqrt{v_{in}^{*2}(\frac{1}{4} + \frac{m^2\pi^2}{h^{*2}} + k^{*2})^2 + 4(\frac{1}{4} + \frac{m^2\pi^2}{h^{*2}} + k^{*2})k^{*2}g^*}}{2(\frac{1}{4} + \frac{m^2\pi^2}{h^{*2}} + k^{*2})} \tag{32}$$
Figure 3 shows the dimensionless dispersion relation for configuration $h^* = 1$, $m =$
1, $g^* = 10$, $g^* = 10$, $v_{in}^*$=0, 0.01, 0.1, 0.3, 0.5. Note the $v_{in}^*$=0.01 line basically
coincides with $v_{in}^*$=0 line. From figure 3 we can see that with the decrease in collision
frequency the linear growth rate increase, and the linear growth rate tends to that of
classical RTI as collision frequency approaches zero.

**4 The linear growth rate of RTI in ionospheric F layer**

In ionospheric F layer $V_A^2/c^2$ is typically very small, the linear growth rate of RTI
should be equation (30). The maximum growth rate is $\gamma = \sqrt{\frac{g}{L} + \frac{v_{in}^2}{4}} - \frac{v_{in}}{2}$. The linear
growth rate calculated by Kelley (2009) is $\gamma_K = \frac{g}{Lv_{in}}$. In figure 4 we plotted the linear
growth rate of RTI as a function of collision frequency. Typically values in ionospheric
F layer g=9.8 m/s², L=20 km and $v_{in}$= 10⁻³-10¹ s⁻¹ were used in the plots. As can be
seen from figure 4, as the collision frequency increases, both γ and $\gamma_K$ decrease rapidly.
When the collision frequency is large, the difference between the growth rates
calculated by the two expressions is small. As the collision frequency decreases, the
difference between the growth rates calculated by the two expressions becomes larger,
with γ tends to a specific value 0.022 and $\gamma_K$ tends to infinity.

**5 Discussion**

The difference between the current calculation and previous calculation is that the
charge conservation equation (Eq. (6)) is used instead of the current continuity equation
during the calculation. Take the divergence of Eq. (2) indicate that when the
displacement current term in Eq. (2) is ignored, the current continuity equation is
automatically satisfied. In general, the displacement current term is neglected because
it's typically much smaller than the **J** and curl **B** term, or because of the requirement of
quasi-neutrality considerations. However, as we can see from Eq. (2) that the $\frac{\partial \boldsymbol{E}}{\partial t}$ term
has the same order of magnitude as **J**, and the changing electric field have significant





effects on the dynamics of the plasma (Vasyliūnas, 2012;Buneman, 1992). Also, studies
show that when Alfvén speed is comparable to or larger than the light speed, the
displacement current cannot be neglected, regardless of the quasi-neutrality
considerations (Vasyliunas, 2005;Song and Lysak, 2006;Boris, 1970), and this is
consistent with our result discussed in 3.2. Time scale analysis showed that everything
involving charge separation happens on time scales of the inverse plasma frequency,
and in such short time scales, the displacement current term cannot be ignored
(Vasyliunas, 2005;Gombosi et al., 2002). The RTI process involves charge
accumulation, in order to be consistent with the physical description the displacement
current term should not be ignored during the calculation of the growth rate of RTI.
Although the RTI in the plasma involves charge accumulation, the contribution of
charge accumulation to RTI growth depends on the ratio of Alfven velocity to the speed
of light. In ionospheric F layer Alfvén speed is typically much smaller than the speed
of light, the contribution of charge accumulation to the growth of RTI can be ignored.
The physical description shown in figure 1 which attribute the growth of RTI to charge
accumulation is inaccurate. A more reasonable physical description of RTI should be
like this (see figure 5): In equilibrium state a net current flows in in the horizontal
direction and the current is proportional to the plasma density. There is thus a
divergence, and charge will pile up on the edges of the small initial perturbation, and
the perturbation electric field try to increase the initial perturbation. However, as
discussed above the contribution of charge accumulation to the growth of RTI can be
neglected. At the same time, when a parcel of heavier plasma is displaced downward
with an equal volume of lighter plasma displaced upwards, the potential energy of the
system decreases, and the process goes on. That is the tendency to decrease the potential
energy of the system is the main driving force for the growth of RTI. The downward or
upward movement of the plasma create the horizontal perturbation electric field and
charge accumulation. The perturbation electric field and charge accumulation is not the
cause, but the result of RTI. The fact that the linear growth rate of RTI in plasma without
collision and the linear growth rate of RTI in neutral fluid are both $\gamma = \sqrt{\frac{g}{L}}$ also
implies that charge accumulation is not the driving force of RTI.

The linear growth rate calculated by Kelley (2009) is $\gamma = \frac{g}{L v_{in}}$, which tends to infinity
when the collision frequency approaches zero, this is physically unreasonable. The
problem in his calculation is that perturbation electric field and current continuity
equation were applied at the same time. Current continuity means no charge
accumulation due to the divergence of the current, and no associated perturbation
electric field. Kelley (2009) also generalized the RTI by include the effects of
background electric field and neutral wind. He think the fundamental destabilizing
source is the current, background electric field and neutral wind create electric current
and affect the linear growth rate of RTI. However, as discussed above the contribution
of charge accumulation to the growth of RTI can be ignored in ionospheric F layer,





background electric field and neutral wind velocity has no effects on the linear growth
rate of RTI. In ionopheric F layer, the maximum growth rate is $\gamma = \sqrt{\frac{g}{L} + \frac{v_{in}^2}{4}} - \frac{v_{in}}{2}$.

## 6 Conclusions

The linear growth rate of RTI was calculated with the standard instability analysis
method. In order to be consistent with the physical description and estimate the
contribution of charge accumulation to the growth of RTI, the charge conservation
equation was used instead of the current continuity equation during the calculation. The
results shows that the contribution of charge accumulation to the growth of RTI is
proportional to the ratio of the Alfvén speed to the light speed. In ionospheric F layer,
Alfvén speed is much smaller than the light speed, the contribution of charge
accumulation to the growth of RTI can be neglected. The physical description of RTI
which consider the charge accumulation as the cause of RTI is inaccurate. In the new
physical description, charge accumulation and the perturbation electric field is not the
cause, but the result of RTI. In ionospheric layer, background electric field and neutral
wind velocity has no effect on the linear growth rate of RTI, the linear growth rate of
RTI in ionospheric F layer is $\gamma = \sqrt{\frac{g}{L} + \frac{v_{in}^2}{4}} - \frac{v_{in}}{2}$,

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



**Figures**

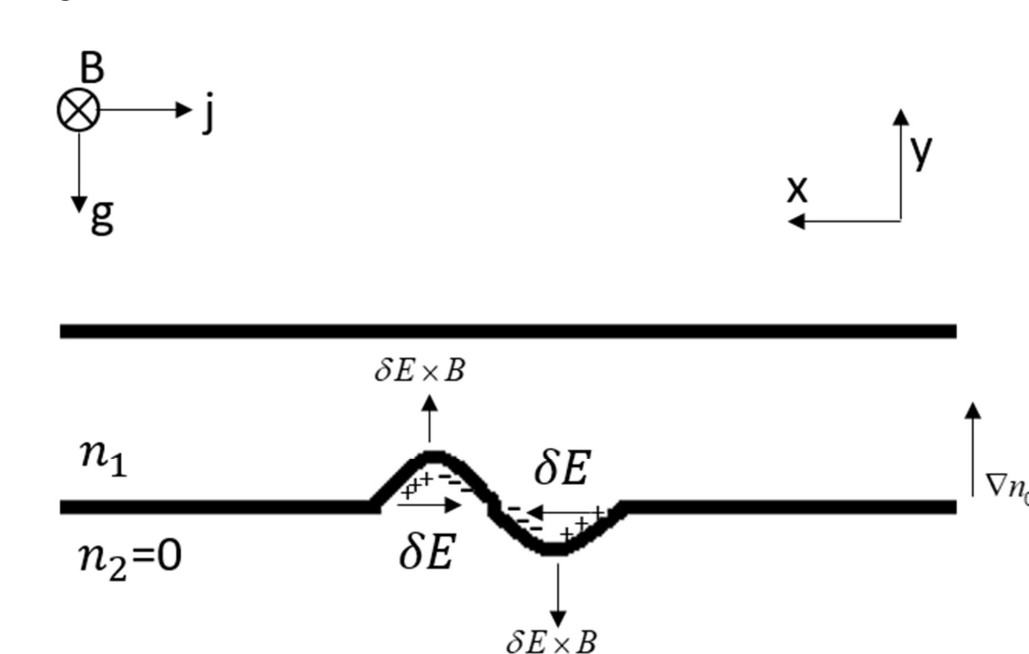


Figure 1. Schematic diagram of the RTI in the equatorial geometry. In this physical
description, charge accumulation and the perturbation electric field is the cause of RTI.





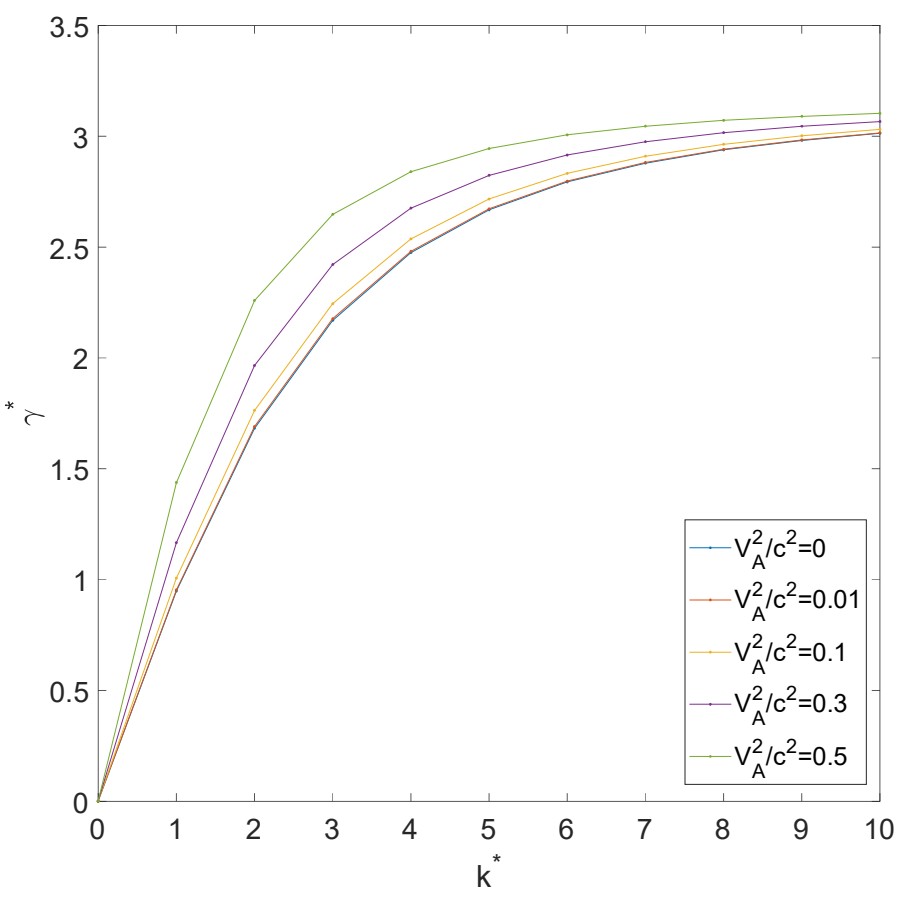


Figure 2. The growth rate of RTI ($\gamma^*$) versus wave number ($k^*$) for different values of
$V_A{}^2/c^2$. $V_A$ and c are the Alfvén speed and light speed, respectively.




.

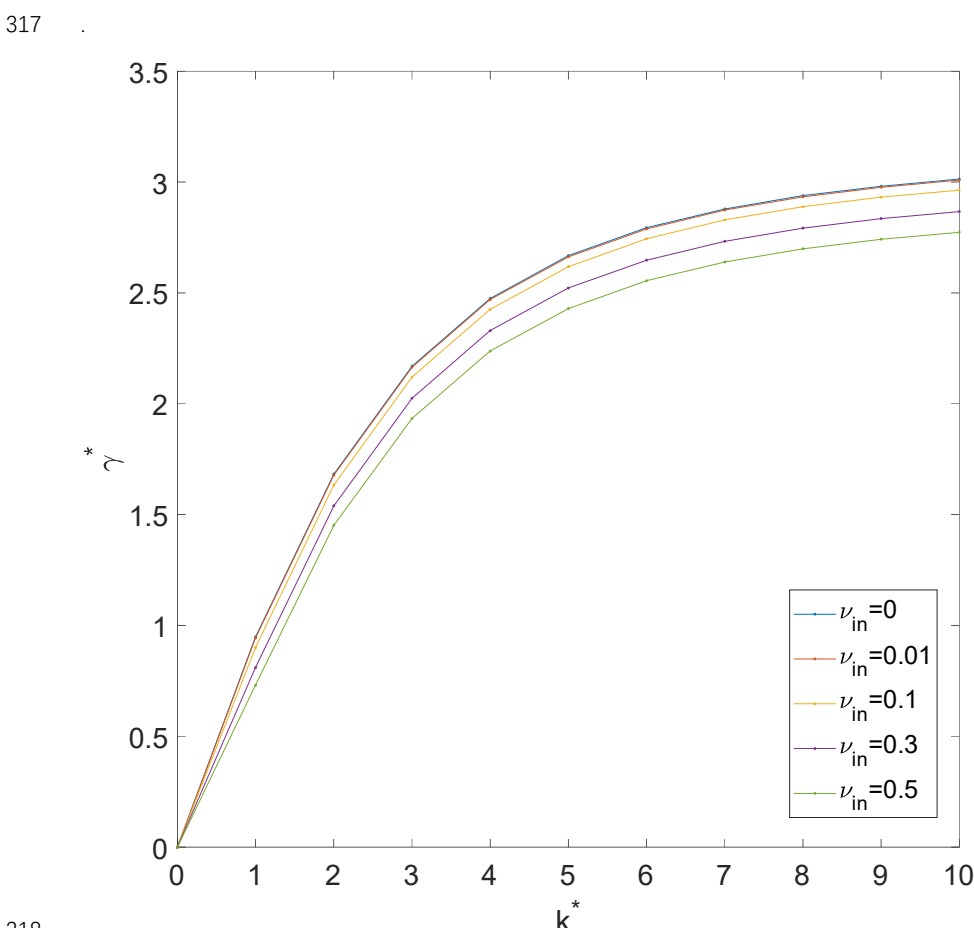


Figure 3. The growth rate of RTI ($\gamma^*$) versus wave number ($k^*$) for different values of

$v_{in}{}^*$.




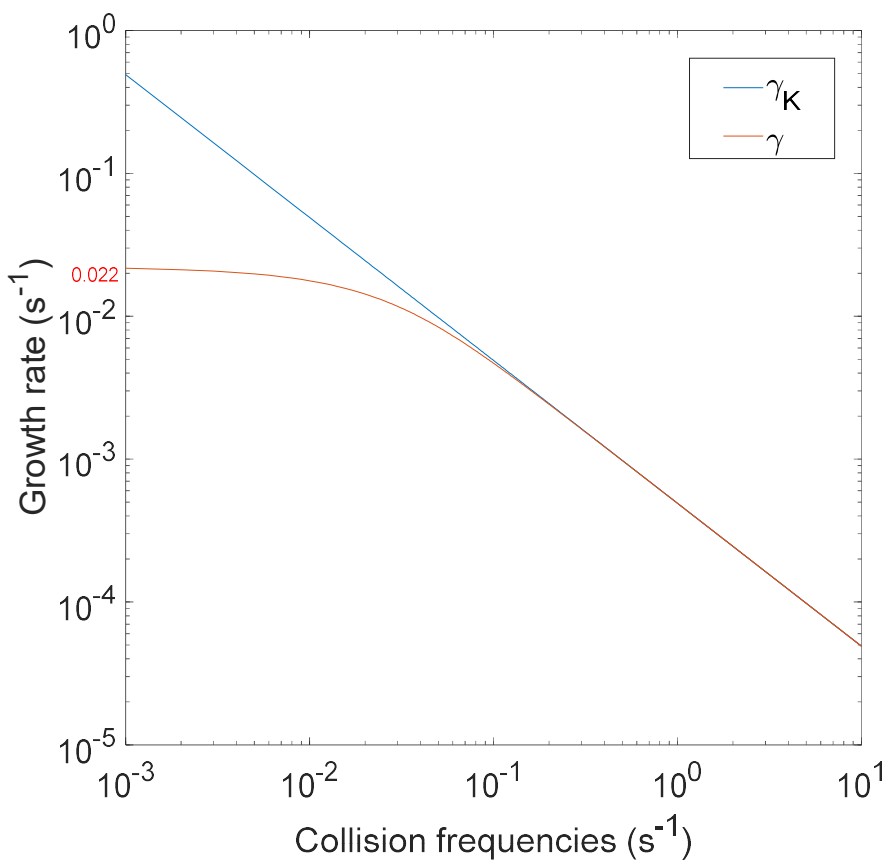


Figure 4. The linear growth rate of RTI as a function of collision frequency. $\gamma =$
$\sqrt{\frac{g}{L} + \frac{v_{in}^2}{4}} - \frac{v_{in}}{2}$, $\gamma_K = \frac{g}{L v_{in}}$.

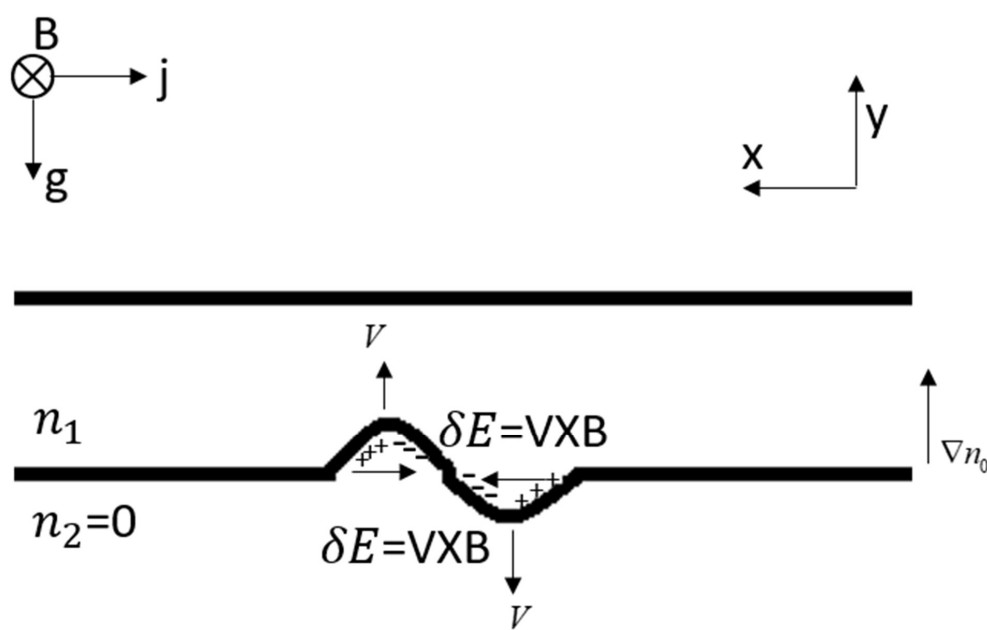


Figure 5. Schematic diagram of the RTI in the equatorial geometry. In this physical
description, charge accumulation and the perturbation electric field is the result of RTI.