# Peer review of "The linear growth rate of Rayleigh-Taylor instability in ionospheric F layer"

_Annales Geophysicae, 2019_

## Referee Comment (RC1) · Anonymous Referee #1 · 26 Mar 2019

This study challenges the standard ionospheric electrodynamics and proposes the new definition of the linear growth rate of the Rayleigh-Taylor instability (RTI) in the equatorial ionosphere. The commonly used linear growth rate of RTI in the equatorial ionosphere is written as $g/(L\nu_{in})$. Author noticed the problem that the growth rate is going to be infinity when $\nu_{in}$ goes to zero, whereas the growth rate of RTI in the collision-less plasma should have finite value. The obtained result seems to connect the theoretical gap between collisional and collision-less plasma naturally and build a seamless instability theory from the ionosphere to magnetosphere. However, the assumption to derive the old expression is usually valid in the ionosphere, and the difference between the old and new growth rate is negligible. Therefore, the title and the conclusion of the paper are misleading. I recommend that author should not

focus on the equatorial ionospheric F layer, but on filling the theoretical gap between collisional and collision-less plasma, which may be an interesting topic.

Author mentioned in page 2 that "when current continuity equation applied, there will be no perturbation electric field due to charge accumulation." It is not correct description. From the Gauss law, $\partial\rho/\partial t + \nabla \cdot J = 0$ is derived. It means $\partial\rho/\partial t = 0$ when the current continuity is satisfied. It does not say $\rho = 0$. Due to very small charge accumulation, electrostatic polarization field is set up. The charge accumulation is so small that the current continuity equation is applied in the electrodynamics in the ionosphere. Authors should estimate quantitatively the amount of charge accumulation produced during the Rayleigh-Taylor instability process. See Chapter 2.3 of Kelley (2009). Very small charge accumulation could produce large electric field. I think the new point in this paper is the inclusion of $\partial E/\partial t$ term in Equation (15). It is very small in the ionosphere, and is going to be important when the ratio of Alfvén speed to the speed of light becomes large.

In order to compare the old and new growth rate intuitively, the new growth rate should be written in the following way.

$$
\gamma \;=\; \frac{\left(\sqrt{\frac{g}{L}+\frac{\nu_{in}^2}{4}}-\frac{\nu_{in}}{2}\right)\left(\sqrt{\frac{g}{L}+\frac{\nu_{in}^2}{4}}+\frac{\nu_{in}}{2}\right)}{\left(\sqrt{\frac{g}{L}+\frac{\nu_{in}^2}{4}}+\frac{\nu_{in}}{2}\right)}
$$

$$
=\; \frac{g}{L}\frac{1}{\sqrt{\frac{g}{L}+\frac{\nu_{in}^2}{4}}+\frac{\nu_{in}}{2}} = \frac{g}{L\nu_{in}}\frac{1}{\sqrt{\frac{g}{L\nu_{in}^2}+\frac{1}{4}}+\frac{1}{2}}
$$

Then it can be easier to understand when the new terms become significant. When $\nu_{in}^2$

is significantly larger than $g/L$, which is usually satisfied in the ionosphere, the growth rate turns to be the traditional expression $g/(L\nu_{in})$. In Figures 2 and 3, the estimated growth rate is plotted with regard to the normalized parameters. What altitude do these parameters correspond to? If the new growth rate should be applied in the ionosphere, substitute the typical values of collision frequency and Alfvén speed of the ionosphere, and show how the growth rate is modified.

---

## Author Comment (AC1) · 2 Apr 2019

**Reply to Referee_1**

I would like to thank the Referee_1 for the questions and suggestions that helped me improve the manuscript. Below I will provide answers to your questions.

Referee 1:

This study challenges the standard ionospheric electrodynamics and proposes the new definition of the linear growth rate of the Rayleigh-Taylor instability (RTI) in the equatorial ionosphere. The commonly used linear growth rate of RTI in the equatorial ionosphere is written as $g/(Lv_{in})$. Author noticed the problem that the growth rate is going to be infinity when $v_{in}$ goes to zero, whereas the growth rate of RTI in the collision-less plasma should have finite value. The obtained result seems to connect the theoretical gap between collisional and collision-less plasma naturally and build a seamless instability theory from the ionosphere to magnetosphere. However, the assumption to derive the old expression is usually valid in the ionosphere, and the difference between the old and new growth rate is negligible. Therefore, the title and the conclusion of the paper are misleading. I recommend that author should not focus on the equatorial ionospheric F layer, but on filling the theoretical gap between collisional and collision-less plasma, which may be an interesting topic.

Reply: I think the assumptions to derive the old expression by Kelley (2009) is physically unreasonable for several reasons. 1. The assumption that $\nabla \cdot J = 0$ and the growth of RTI is due to charge accumulation is contradicted. From the Gauss law, $\partial\rho/\partial t + \nabla \cdot J = 0$ we know that when $\nabla \cdot J = 0$, $\partial\rho/\partial t = 0$. It does not say $\rho = 0$, however, in the initial state of RTI $\rho = 0$, $\nabla \cdot J = 0$ means $\rho$ will always equals to zero. 2. The assumption that the growth of RTI is due to charge accumulation is invalid. It is shown quantitatively in this manuscript that the contribution of charge accumulation to the growth of RTI is ignorable. Qualitatively, the divergence of current create charge accumulation, the associated electric field tries to amplify the initial perturbation. At the same time, the electric field drives a current with reduce the charge accumulation. The net result may be charge accumulation is so small that the contribution to the growth of RTI can be neglected. 3. Base on the above assumptions, the full expression of the linear growth rate in the ionosphere F region calculated by Kelley (2009) (Chapter 4.2) is $\gamma = g/(Lv_{in})$, it did not say how the growth rate changes with the wavenumber k. In real circumstances, the linear growth rate is a function of the wavenumber k.

Since the old growth rate is not a function of the wavenumber k, in the manuscript I compared the old growth rate with the maximum of the new growth rate. As shown in figure 4 of the manuscript, the different between the old and maximum of the new growth rate is negligible when $v_{in}$ is in the range $[10^{-1}-10^{1}]$, the corresponding altitude

range is [200-500] km (Kelley, 2009). When $v_{in}$ is in the range $[10^{-2}-10^{-1}]$, the corresponding altitude range is [500-700] km, the difference is large. In real circumstances, when the wavenumber k is small, the difference between the old and new growth rate maybe large.

When studying the linear growth rate of plasma in magnetic field, I found the deriving process of the linear growth rate of RTI in ionosphere by Kelley (2009) is quite different form the normal ways. Generally, the linear growth rate for collision-less plasma ($v_{in} = 0$) is derived by linearize the momentum equation, and later the boundary conditions are required to solve the related equations, the resultant linear growth rate is a function of the wavenumber k. However, the deriving process of the linear growth rate of RTI in ionosphere ($v_{in} > 0$) was done by linearize the particle continuity equation and no boundary conditions was used, the resultant linear growth rate is not a function of the wavenumber k. It should be noted that in previous deriving process of linear growth rate for collision-less plasma in magnetic field, $\nabla \cdot J = 0$ was used and the resultant growth rate is $\gamma = \sqrt{\frac{g}{L}}$ , which is the same as that of neutral fluid. The process to derive the linear growth rate in collisional plasma is simply add the collisional term in the related equations. The difference between the old and new linear growth rate is negligible in low altitude ionosphere F region. However, during the old deriving process, the only constrain is $v_{in} > 0$, as long as $v_{in}$ is not zero, the result should apply. However, the old linear growth rate, tends to infinite, when $v_{in}$ is very small, which is physically unacceptably. The new linear growth rate when $v_{in}$ tends to zero, automatically reduced to that of collision-less may indicate that the physics between collisional and collision-less plasma RTI is the same. In conclusion, in this manuscript, I give an accurate expression of the linear growth rate of RTI, and shows that the growth of RTI is not due to the charge accumulation.

Author mentioned in page 2 that "when current continuity equation applied, there will be no perturbation electric field due to charge accumulation." It is not correct description. From the Gauss law, $\partial\rho/\partial t + \nabla \cdot J = 0$ is derived. It means $\partial\rho/\partial t = 0$ when the current continuity is satisfied. It does not say $\rho = 0$. Due to very small charge accumulation, electrostatic polarization field is set up. The charge accumulation is so small that the current continuity equation is applied in the electrodynamics in the ionosphere. Authors should estimate quantitatively the amount of charge accumulation produced during the Rayleigh-Taylor instability process. See Chapter 2.3 of Kelley (2009). Very small charge accumulation could produce large electric field. I think the new point in this paper is the inclusion of $\partial E/\partial t$ term in Equation (15). It is very small in the ionosphere, and is going to be important when the ratio of Alfvén speed to the speed of light becomes large.

Reply: When current continuity is satisfied, $\partial\rho/\partial t = 0$. Yes, it does not say $\rho = 0$. However, $\partial\rho/\partial t = 0$ means $\rho$ remains constant. If in the initial state $\rho = 0$, $\rho$ will be constantly zero and there will be no charge accumulation. If in the initial state

ρ = C, where C is some constant greater than zero, ρ will equals to C in later times. So, we can say that when initially ρ = 0, when current continuity equation satisfied, there will be no perturbation electric field due to charge accumulation. When initially ρ = C, when current continuity equation satisfied, there will be no perturbation electric field due to additional charge accumulation. In the description of Rayleigh-Taylor instability (RTI) in equatorial ionosphere by Kelley (2009) (Chapter 4.2 "Since the current is in the g×B direction, which is strictly horizontal, $J_x$ will be large when $n$ is large and small when $n$ is small. There is thus a divergence, and charge will pile up on the edges of the small initial perturbation."), in the initial state ρ = 0 and later ρ increases, which means $\partial\rho/\partial t \neq 0$ or equivalently $\nabla \cdot J \neq 0$ during the growth of RTI. If one attribute the growth of RTI in equatorial ionosphere to charge accumulation such as Kelley (2009), or want to study the contribution of charge accumulation to the growth of RTI, $\nabla \cdot J = 0$ should not be used during the calculation of the linear growth rate of RTI.

It is usually accepted that when $\nabla \cdot J \neq 0$, charge density will creates an electric field that forces the divergence to zero, so $\nabla \cdot J = 0$ was used. However, $\nabla \cdot J = 0$ means $\nabla \cdot J$ is strictly equals to zero, which indicate that ρ will remain constant. Which is not the case in most circumstances. In steady state $\nabla \cdot J = 0$ can be applied, in unsteady state the constraint $\nabla \cdot J = 0$ is too strict. In unsteady state the electric field due to charge accumulation tries to force the divergence to zero but failed, the net effect is to keep $\nabla \cdot J$ small but not strictly equals to zero. The RTI process is obviously not in a steady state, so $\partial\rho/\partial t + \nabla \cdot J = 0$ should be used. However, as shown in the manuscript, the process of RTI involves charge accumulation, but the effect of charge accumulation to the growth of the RTI is negligible in equatorial ionosphere. So using $\nabla \cdot J = 0$ when deriving the linear growth rate of RTI in the equatorial ionosphere is safe, but simultaneously using the current continuity equation and perturbation electric field equation is inaccurate. Also, the physical description that the growth of RTI is due to charge accumulation is inaccurate. When $\nabla \cdot J = 0$ is used, the contribution of charge accumulation to the growth of RTI is totally neglected.

The inclusion of $\partial E/\partial t$ term in Equation (15) is possible is due to that fact that $\nabla \cdot J$ is not strictly zero. If $\nabla \cdot J = 0$, take the divergence of equation $\frac{\partial E}{\partial t} = -4\pi J + c\nabla \times B$ we get $\nabla \cdot \frac{\partial E}{\partial t} = 0$, if E is created by charge accumulation, $\frac{\partial E}{\partial t} = 0$. Or if $\frac{\partial E}{\partial t} = 0$, take the divergence of $\frac{\partial E}{\partial t} = -4\pi J + c\nabla \times B$, we get $\nabla \cdot J = 0$.

During RTI process, due to the divergence of the current density, charge pile up on the edges of the small initial perturbation which create perturbation electric field, the electric field tries to amplify the initial small perturbation, and at the same time, the perturbation electric field tries to forces the divergence of the current density to zero. It seems that there will be not much charge accumulation and the effect of the associated

electric field is limited. It is hard to estimate quantitatively the amount of charge accumulation produced during the RTI process. However, in the manuscript I estimate quantitatively the contribution of the charge accumulation to the growth of RTI. The results shows that the contribution of the charge accumulation to the growth of RTI is related to the ratio of Alfvén speed to the light speed. In equatorial ionosphere, this ratio is very small, the contribution of charge accumulation to the growth of RTI can be neglected. Using $\nabla \cdot J = 0$ when deriving the linear growth rate in equatorial ionosphere is safe, but deriving process is questionable and the description of the RTI process by Kelley (2009) was inaccurate.

In order to compare the old and new growth rate intuitively, the new growth rate should be written in the following way.

$$\gamma = \frac{\left(\sqrt{\frac{g}{L}+\frac{v_{in}^2}{4}}-\frac{v_{in}}{2}\right)\left(\sqrt{\frac{g}{L}+\frac{v_{in}^2}{4}}+\frac{v_{in}}{2}\right)}{\left(\sqrt{\frac{g}{L}+\frac{v_{in}^2}{4}}+\frac{v_{in}}{2}\right)}$$

$$= \frac{g}{L}\frac{1}{\sqrt{\frac{g}{L}+\frac{v_{in}^2}{4}}+\frac{v_{in}}{2}} = \frac{g}{Lv_{in}}\frac{1}{\sqrt{\frac{g}{Lv_{in}^2}+\frac{1}{4}}+\frac{1}{2}}$$

Then it can be easier to understand when the new terms become significant. When $v_{in}^2$ is significantly larger than g/L, which is usually satisfied in the ionosphere, the growth rate turns to be the traditional expression $g/(Lv_{in})$. In Figures 2 and 3, the estimated growth rate is plotted with regard to the normalized parameters. What altitude do these parameters correspond to? If the new growth rate should be applied in the ionosphere, substitute the typical values of collision frequency and Alfvén speed of the ionosphere, and show how the growth rate is modified.

Reply: Yes, write the new growth rate in the above form is intriguing, I will use the above form in the manuscript. In Figures 2 and 3, I just want to show how the growth rate of changes with the ratio of Alfvén speed to light speed and the collision frequency. In Figure 4 I showed the variation of the maximum growth rate with collision frequency with typical values in the ionosphere F layer. See from Figure 1, $v_{in}$ in the range of $10^{-3}$-$10^1$, the corresponding altitude is around 200- 900 km. Seen from figure 4 in the manuscript only in the low altitude F region the difference is negligible. The Alfvén speed is too small in the ionosphere, also, even if Alfvén speed is large, for the maximum growth rate (the wavenumber tends to infinity), the Alfvén speed term will be vanished, and the effect of charge accumulation is negligible.

[Figure]

**Figure 2.3** Typical electron neutral plus electron ion collision frequency along with the ion-neutral collision frequency at a high sunspot number.

Figure 1. Figure 2.3 in Chapter 2.2 of Kelley (2009)

Reference

Kelley, M. C.: The Earth's ionosphere: plasma physics and electrodynamics, Academic press, 2009.

Sincerely,

Kangkang Liu

---

## Referee Comment (RC2) · Anonymous Referee #2 · 29 Apr 2019

General comments:

Presented in this manuscript is a new formulation of the linear growth rate for the Rayleigh-Taylor plasma instability (RTI) that generates Equatorial Plasma Bubbles (EPBs) in the equatorial ionospheric F layer. Different from previous formulations, notably that by Kelley (2009), this formulation uses charge conservation as opposed to current continuity. The author argues that there are issues in applying both the current continuity equation and the perturbation electric fields, as done in previous formulations. This results in a significantly different interpretation of the RTI; in particular, that both the charge accumulation and the perturbation electric fields are not the causes of the RTI, but are the result of it, as argued in this manuscript. The author also concludes that both the background electric field and the neutral wind do not affect the RTI. This

manuscript presents an interesting perspective on the RTI formulation and, in the opinion of this reviewer, does constitute an advancement in the theoretical understanding of the RTI. However, there are a few, mostly minor, issues (noted below) that the author should consider prior to being fully accepted for publication.

Specific comments:

1. On the effect of the background electric field and the neutral wind on the RTI: The conclusion that the background electric field and neutral wind do not impact the RTI is not substantiated in my view. Perhaps I did not follow the derivation presented in section 2 well, but it appears to me that the background electric and neutral wind were not considered. In particular, line 94 states that the background electric field was set to zero, and the neutral wind, which is present in equation (1), is no longer present following the linearization in equation (9) and it's not clear to me why this is the case; was there a reference frame change to that of the neutral wind or was it also set to zero? In either case, the author should clarify this point and either justify this conclusion (by better explaining the derivation in section 2) or remove this conclusion from the manuscript.

2. Exponential decay of plasma density with altitude: Equation (18) states that the initial plasma density has an exponential dependence on y. How does this assumption impact the updated description of the RTI shown in figure 5, where the plasma density is represented as a step function from n2=0 to n1? Should this formula for the plasma density be included in this schematic; i.e., should this formula be included for n1 to further complete this description? Further, is this description for L the same as other formulations; i.e., gradient scale length L in Kelley (2009)?

3. General readability of derivation: It is suggested that the author attempt to improve the readability of the derivation in section 2; e.g., in lines 116-120, there are references the "the above relation" and it's not clear to the reader which formula is being discussed, and some of the English is not entirely clear. For example, "the exact relation between

E and V . . . is not simply cE+VxB = 0. . ." yet this relation is used to obtain equation (16). This section needs to be clarified.

4. Local RTI versus flux-tube integrated RTI: It is clear to this reader that the formulation in the current manuscript is focused on the "local" RTI, as opposed to the flux-tube integrated RTI, which was derived by Sultan (1996), which is arguably more "accurate" than any formulation of the local RTI, which includes only a 2-D description of the phenomenon that ignores aspects like interhemispherical asymmetries in physical parameters. While requesting the author to expand their formulation from 2-D to 3-D is clearly outside of the scope of this work, it is suggested to the author to include some mention of this previous work in their introduction and to include some comments on the differences between these approaches in their manuscript. Such additions would help readers better place this work in the context of previous theoretical works.

Technical corrections:

1. Line 14: "calculations" instead of "calculation"

2. Line 24: "this ratio" instead of "the ratio"

3. Lines 25-26: "previous physical description. . . is wrong" is quite strong language. I would suggest that perhaps "is inaccurate" is a better choice.

4. Line 45: Remove second "in"

5. Line 53: I think the author means "From" from "Form"

6. Line 57: "contradicts" not "contradict"

7. Lines 60-61: "equatorial plasma bubble" and "EPB" to plural

8. Line 63: "when the current continuity is applied"

9. Line 65: "growth rate he calculated" I suggest removing personal references from the manuscript

10. Line 71: "RTI are"

11. Line 73: "F layer is depicted"

12. Lines 116 and 118: is "v" supposed to be capitalized?

13. Line 159: "g=10" appears twice here

14. Line 167: This language needs to be smoothened.

15. Line 169: "As k increases.."

16. Line 183: The term "gamma subscript K" for Kelley's growth rate formula appears for the first time here, but does not appear to be used throughout; e.g., line 232 and 59-60. I suggest making this consistent throughout the manuscript

17. Line 194: "Taking the divergence"

18. Line 199: "field has significant"

19. Line 204: "discussed in section 3.2"

20. Lines 207-208: Should this sentence be two separate sentences? "...accumulation. In order..."

21. Line 237: "by including the effects"

22. Line 238: Sentence beginning with "He think" needs to be reworded; also the previous sentence needs a period. Further, remove personal references.

---

## Author Comment (AC2) · 24 May 2019

**Reply to Referee_2**

I would like to thank the Referee_2 for the questions and suggestions that helped me improve the manuscript. Below I will provide answers to your questions.

Referee_2:

Specific comments:

1. On the effect of the background electric field and the neutral wind on the RTI: The conclusion that the background electric field and neutral wind do not impact the RTI is not substantiated in my view. Perhaps I did not follow the derivation presented in section 2 well, but it appears to me that the background electric and neutral wind were not considered. In particular, line 94 states that the background electric field was set to zero, and the neutral wind, which is present in equation (1), is no longer present following the linearization in equation (9) and it's not clear to me why this is the case; was there a reference frame change to that of the neutral wind or was it also set to zero? In either case, the author should clarify this point and either justify this conclusion (by better explaining the derivation in section 2) or remove this conclusion from the manuscript.

Reply: For the sake of simplicity, the background electric field $\mathbf{E}$ and neutral wind $\mathbf{V}_n$ are set to zero during the derivation process, and $\mathbf{E}$ and $\mathbf{V}_n$ did not appear in the expression of linear growth rate of RTI. It should be noted that $\mathbf{V}_n$ and $\mathbf{E}$ also do not appear in the expression of the linear growth rate of RTI calculated by Kelley (2009, Chapter 4.2.1). When deriving the generalized linear growth rate of RTI, Kelley (2009, Chapter 4.2.2) thought that the fundamental destabilizing source of RTI is the current and the background electric field and neutral wind will drive a current, $\mathbf{J} = \boldsymbol{\sigma} \cdot \mathbf{E}'$ where $\mathbf{E}' = \mathbf{E_0} + \mathbf{U} \times \mathbf{B}$. However, as discussed in the manuscript that the effect of charge accumulation due to the divergence of the current can be neglected, and base on the above statement, it was considered that the background electric field and neutral wind do not impact the linear growth rate of RTI. Since, the fundamental destabilizing source of RTI was gravity, the generalizing process should be related to gravity-like forces.

In the deriving process in the manuscript, in the initial equilibrium state, the plasma is assumed stationary, that is $\mathbf{V^0} = 0$, $\mathbf{V} = \mathbf{V^1}$, and $\mathbf{V}_n = 0$. Since electric filed did not appear in the momentum equation, electric field will not appear in the expression of the linear growth rate of RTI. If a constant background electric field $\mathbf{E^0}$ is present, the plasma will moves with a constant velocity $\mathbf{V^0}$, with $\mathbf{V^0}$ and $\mathbf{E^0}$ related by the expression $c\mathbf{E^0} + \mathbf{V^0} \times \mathbf{B^0} = 0$. To study the effect of background electric field and neutral wind on the linear growth rate of RTI, the initial equilibrium state condition should be changed. The initial equilibrium state condition is that the plasma moves with $\mathbf{V^0}$, $\mathbf{V} = \mathbf{V^0} + \mathbf{V^1}$ and $\mathbf{V}_n \neq 0$.

The momentum equation is

$$\frac{\partial(\rho V)}{\partial t} = \frac{1}{c}\boldsymbol{J} \times \boldsymbol{B} + \rho\boldsymbol{g} - \nabla p - \rho v_{in}(\boldsymbol{V} - \boldsymbol{V}_n) \tag{1}$$

In initial equilibrium state, the left term Eq.(1) vanish, the $\boldsymbol{J} \times \boldsymbol{B}$ force and other forces are balanced, we have

$$\frac{1}{c}\boldsymbol{J}^0 \times \boldsymbol{B}^0 + \rho^0\boldsymbol{g} - \nabla p^0 - \rho^0 v_{in}(\boldsymbol{V}^0 - \boldsymbol{V}_n) = 0 \tag{2}$$

Linearizing Eq.(1), we have

$$\rho^0\frac{\partial \boldsymbol{V}^1}{\partial t} + \boldsymbol{V}^0\frac{\partial \rho^1}{\partial t} = \frac{1}{c}\boldsymbol{J}^0 \times \boldsymbol{B}^1 + \frac{1}{c}\boldsymbol{J}^1 \times \boldsymbol{B}^0 + \frac{1}{c}\boldsymbol{J}^0 \times \boldsymbol{B}^0 + \rho^1\boldsymbol{g} + \rho^0\boldsymbol{g} - \nabla p^1 - \nabla p^0 - \rho^0 v_{in}\boldsymbol{V}^1 - \rho^0 v_{in}\boldsymbol{V}^0 - \rho^1 v_{in}\boldsymbol{V}^0 - \rho^1 v_{in}(-\boldsymbol{V}_n) - \rho^0 v_{in}(-\boldsymbol{V}_n) \tag{3}$$

Substituting Eq.(2) into Eq.(3) we get

$$\rho^0\frac{\partial \boldsymbol{V}^1}{\partial t} + \boldsymbol{V}^0\frac{\partial \rho^1}{\partial t} = \frac{1}{c}\boldsymbol{J}^0 \times \boldsymbol{B}^1 + \frac{1}{c}\boldsymbol{J}^1 \times \boldsymbol{B}^0 + \rho^1\boldsymbol{g} - \nabla p^1 - \rho^0 v_{in}\boldsymbol{V}^1 - \rho^1 v_{in}\boldsymbol{V}^0 - \rho^1 v_{in}(-\boldsymbol{V}_n) \tag{4}$$

Assuming perturbations in the form

$$\psi \propto \psi(y)e^{i(kx-\omega t)} \tag{5}$$

where $\omega$ is the frequency of the perturbation, $k$ is the wave number.
For simplicity, assume $\boldsymbol{V}^0 = (0, V^0, 0)$, and $\boldsymbol{V}_n = (V_n, 0, 0)$. Which correspond to zonal background electric field and zonal neutral wind in the equatorial ionospheric F layer.
$\boldsymbol{z} \cdot \nabla \times$ Eq. (4) yields

$$-i\omega(ik\rho^0 V_y - \frac{\partial}{\partial y}(\rho^0 V_x) + ik\rho^1 V^0) = -ik\rho^1 g - \frac{1}{c}(\nabla \cdot \boldsymbol{J}^1)B^0 - v_{in}(ik\rho^0 V_y - \frac{\partial}{\partial y}(\rho^0 V_x) + ik\rho^1 V^0) \tag{6}$$

Ignore the effect of charge accumulation and following the deriving process in the manuscript, we can get the maximum linear growth rate

$$\gamma = \sqrt{\frac{g+v_{in}V^0}{L} + \frac{v_{in}^2}{4}} - \frac{v_{in}}{2} \tag{7}$$

From above expression, we can see that g was replaced by $g^* = g + v_{in}V^0$. When y component of $\boldsymbol{V}_n$ is zero and $\boldsymbol{V}^0$ has only x component, $v_{in}V^0$ is a gravity-like force. Since, the fundamental destabilizing source of RTI was gravity, the generalized maximum linear growth rate should be

$$\gamma = \sqrt{\frac{g+v_{in}(V_y^0-V_{ny})}{L} + \frac{v_{in}^2}{4}} - \frac{v_{in}}{2} \tag{8}$$

$\boldsymbol{V}^0$ is related with background electric field $\boldsymbol{E}^0$ with $c\boldsymbol{E}^0 + \boldsymbol{V}^0 \times \boldsymbol{B}^0 = 0$, the generalized linear growth rate now is

$$\gamma = \sqrt{\frac{g+v_{in}(\frac{cE_x^0}{B^0}-V_{ny})}{L} + \frac{v_{in}^2}{4}} - \frac{v_{in}}{2} \tag{9}$$

The generalized linear growth rate is related to background electric field in x direction and neutral wind velocity in y direction. In equatorial ionospheric F region, zonal neutral wind velocity has no effect on the linear growth rate of RTI, eastward (westward)

background electric field increase (decrease) the linear growth rate of RTI.

2. Exponential decay of plasma density with altitude: Equation (18) states that the initial plasma density has an exponential dependence on y. How does this assumption impact the updated description of the RTI shown in figure 5, where the plasma density is represented as a step function from n2=0 to n1? Should this formula for the plasma density be included in this schematic; i.e., should this formula be included for n1 to further complete this description? Further, is this description for L the same as other formulations; i.e., gradient scale length L in Kelley (2009)?

Reply: The linear growth rate of RTI for both plasma density that has an exponential dependence on y and plasma density that is a step function in y direction has been derived previously. Goldston and Rutherford (1995, Chapter 19.1) calculated the linear growth rate of RTI for plasma density that has an exponential dependence on y. The linear growth rate was

$$\gamma = (\frac{g}{L}\frac{h^2 k^2}{h^2 k^2 + m^2 \pi^2 + h^2/4L^2})^{\frac{1}{2}} \tag{10}$$

The linear growth rate for plasma density that is a step function in y direction previously calculated by Chandrasekhar (2013) was

$$\gamma = \left(g\frac{\rho_2 - \rho_1}{\rho_2 + \rho_1}k\right)^{\frac{1}{2}} \tag{11}$$

Where $\rho_1$ and $\rho_2$ are plasma density below and above the interface.

Following the deriving process of Chandrasekhar (2013), it can be easily shown that the linear growth rate for plasma density that is a step function in y direction is

$$\gamma = \sqrt{g\frac{\rho_2 - \rho_1}{\rho_2 + \rho_1}k + \frac{v_{in}^2}{4}} - \frac{v_{in}}{2} \tag{12}$$

The definition of density gradient scale length in Kelley (2009) is

$$L = \left(\frac{1}{\rho_0}\frac{\partial \rho_0}{\partial y}\right)^{-1} \tag{13}$$

The plasma density profile in the manuscript is

$$\rho^0(y) = \rho^0 e^{\frac{y}{L}} \tag{14}$$

It can be easily see that L in the density profile is gradient scale length the same as that in Kelley (2009).

3. General readability of derivation: It is suggested that the author attempt to improve the readability of the derivation in section 2; e.g., in lines 116-120, there are references the "the above relation" and it's not clear to the reader which formula is being discussed, and some of the English is not entirely clear. For example, "the exact relation between E and V … is not simply cE+VxB = 0…" yet this relation is used to obtain equation (16). This section needs to be clarified.

Reply: I rewrited this section." The exact relation between E and v in collisional plasma is not simply $cE + V \times B = 0$ (Vasyliunas, 2005). In collisional plasma, with a given E, the plasma will moves with velocity small than the velocity calculated from the relation $cE + V \times B = 0$. For simplicity we will use the relation $cE + V \times B = 0$ and note that the contribution of charge accumulation to the growth of RTI is maximized. From the relation $cE + V \times B = 0$ we get

$$E_x = -\frac{1}{c}V_y B^0 \qquad\qquad (16)\text{ ``}$$

4. Local RTI versus flux-tube integrated RTI: It is clear to this reader that the formulation in the current manuscript is focused on the "local" RTI, as opposed to the flux-tube integrated RTI, which was derived by Sultan (1996), which is arguably more "accurate" than any formulation of the local RTI, which includes only a 2-D description of the phenomenon that ignores aspects like interhemispherical asymmetries in physical parameters. While requesting the author to expand their formulation from 2-D to 3-D is clearly outside of the scope of this work, it is suggested to the author to include some mention of this previous work in their introduction and to include some comments on the differences between these approaches in their manuscript. Such additions would help readers better place this work in the context of previous theoretical works.

Reply: It should be noted that the work by Sultan (1996) was mentioned in the introduction (line 53). I added a paragraph in the discussion. " It should be noted that only the linear growth rate of local RTI was discussed up to now. However, in real circumstances, the linear growth rate of flux-tube integrated RTI should be calculated. Sultan (1996) calculated the linear growth rate of flux-tube integrated RTI, however, the derivation is based on the old physical description in which the fundamental destabilizing source of RTI is the current. The linear growth rate tends to infinity when the collision frequency approaches zero. A more accurate linear growth rate of flux-tube integrated RTI base on the new physical description should be calculated. "

Technical corrections:

1. Line 14: "calculations" instead of "calculation"
Reply: Done.
2. Line 24: "this ratio" instead of "the ratio"
Reply: Done
3. Lines 25-26: "previous physical description: : : is wrong" is quite strong language. I would suggest that perhaps "is inaccurate" is a better choice.
Reply: Done.
4. Line 45: Remove second "in"
Reply: Done
5. Line 53: I think the author means "From" from "Form"
Reply: Done.
6. Line 57: "contradicts" not "contradict"
Reply: Done
7. Lines 60-61: "equatorial plasma bubble" and "EPB" to plural
Reply: Done
8. Line 63: "when the current continuity is applied"
Reply: Done.
9. Line 65: "growth rate he calculated" I suggest removing personal references from the manuscript
Reply: Done

10. Line 71: "RTI are"

Reply: Done

11. Line 73: "F layer is depicted"

Reply: Done

12. Lines 116 and 118: is "v" supposed to be capitalized?

Reply: Done

13. Line 159: "g=10" appears twice here

Reply: Done.

14. Line 167: This language needs to be smoothened.

Reply: This sentence was smoothened. "Ignore the $V_A{}^2/c^2$ term in Eq.(25), the growth rate is"

15. Line 169: "As k increases.."

Reply: Done

16. Line 183: The term "gamma subscript K" for Kelley's growth rate formula appears for the first time here, but does not appear to be used throughout; e.g., line 232 and 59-60. I suggest making this consistent throughout the manuscript

Reply: Done.

17. Line 194: "Taking the divergence"

Reply: Done.

18. Line 199: "field has significant"

Reply: Done.

19. Line 204: "discussed in section 3.2"

Reply: Done.

20. Lines 207-208: Should this sentence be two separate sentences? ": : :accumulation. In order: : :"

Reply: Done.

21. Line 237: "by including the effects"

Reply: Done.

22. Line 238: Sentence beginning with "He think" needs to be reworded; also theprevious sentence needs a period. Further, remove personal references.

Reply: This sentence was reworded. "When deriving the generalized linear growth rate of RTI, Kelley (2009) thought that the fundamental destabilizing source of RTI is the current, the background electric field and neutral wind will drive a current, $\boldsymbol{J} = \boldsymbol{\sigma} \cdot \boldsymbol{E'}$ where $\boldsymbol{E'} = \boldsymbol{E_0} + \boldsymbol{U} \times \boldsymbol{B}$, and contribute to the growth of RTI."

Reference

Kelley, M. C.: The Earth's ionosphere: plasma physics and electrodynamics, Academic press, 2009.
Goldston, R. J., & Rutherford, P. H. (1995). Introduction to plasma physics. CRC Press.
Chandrasekhar, S.: Hydrodynamic and Hydromagnetic Stability, Dover Publications., 2013.

Sincerely,

Kangkang Liu